# Force Generation on the Hallux Is More Affected by the Ankle Joint Angle than the Lesser Toes: An In Vivo Human Study

**DOI:** 10.3390/biology10010048

**Published:** 2021-01-12

**Authors:** Junya Saeki, Soichiro Iwanuma, Suguru Torii

**Affiliations:** 1Faculty of Sport Sciences, Waseda University, 2-579-15 Mikajima, Tokorozawa, Saitama 359-1192, Japan; shunto@waseda.jp; 2Japan Society for the Promotion of Science, 5-3-1 Kojimachi, Chiyoda-ku, Tokyo 102-0083, Japan; 3Faculty of Education & Human Sciences, Teikyo University of Science, 2-2-1 Senjusakuragi, Adachi-ku, Tokyo 120-0045, Japan; s-iwanuma@ntu.ac.jp

**Keywords:** toe flexor muscle strength, intrinsic foot muscle, extrinsic foot muscle, torque-angle relationships, toe grip strength

## Abstract

**Simple Summary:**

This study clarified the difference in force generation characteristics on the hallux and lesser toes. The maximal generated torque on the hallux at the dorsiflexed position of the ankle was higher than that at the plantar-flexion position of the ankle. However, no significant difference existed between the maximal generated torque on the lesser toes at any ankle position. The present study suggested that the force generation characteristic on the hallux is more affected by the ankle joint angle than the lesser toes.

**Abstract:**

The structure of the first toe is independent of that of the other toes, while the functional difference remains unclear. The purpose of this study was to investigate the difference in the force generation characteristics between the plantar-flexion of the first and second–fifth metatarsophalangeal joints (MTPJs) by comparing the maximal voluntary plantar-flexion torques (MVC torque) at different MTPJs and ankle positions. The MVC torques of the first and second–fifth MTPJs were measured at 0°, 15°, 30°, and 45° dorsiflexed positions of the MTPJs, and at 20° plantar-flexed, neutral, and 20° dorsiflexed positions of the ankle. Two-way repeated measures analyses of variance with Holm’s multiple comparison test (MTPJ position × ankle position) were performed. When the MTPJ was dorsiflexed at 0°, 15°, and 30°, the MVC torque of the first MTPJ when the ankle was dorsiflexed at 20° was higher than that when the ankle was plantar-flexed at 20°. However, the ankle position had no significant effect on the MVC torque of the second–fifth MTPJ. Thus, the MVC torque of the first MTPJ was more affected by the ankle position than the second–fifth MTPJs.

## 1. Introduction

Toe flexor muscles, which are activated during walking, include the intrinsic and extrinsic foot muscles [1]. Extrinsic and intrinsic muscles act as global movers and core stabilizers of the foot, respectively [2,3], and strength exercise of the toe flexor muscle can correct the foot kinematics during gait with pes planus [4]. In addition, patients with pes planus have greater extrinsic muscle size and smaller intrinsic muscle size [5]. This suggests that the intrinsic and extrinsic muscles have independent function.

The capacity generated by a muscle is altered by its length [6]. Regarding toe flexor muscles, the length of the extrinsic muscles is altered by the metatarsophalangeal joint (MTPJ) as well as the ankle joint angle [7]. Thus, the plantar-flexion torque of the MTPJ is influenced by the force-length relationship between the intrinsic and extrinsic muscles of the foot when the MTPJ angle changes and by the relationship with the extrinsic muscles when the ankle joint angle changes. In the case of the ankle plantar flexor muscle, the soleus muscle, which is a monoarticular muscle, is easily activated when the knee is flexed where the biarticular muscle is slacked [8]. These studies indicate that changing the length of the extrinsic muscles may affect the selective force generation of the intrinsic muscles. Therefore, investigating the difference in muscle strength at various positions in the two joints can allow us to determine how much intrinsic and extrinsic muscles contribute to force generation.

A previous study reported the relationship between the maximal voluntary contraction (MVC) torque and joint angle (torque-angle relationship) of an MTPJ for all toes [9]. However, the structure of the first toe (hallux) is independent of that of the other toes and is moved using different muscles in most primates, including humans [10]. Our previous study reported that runners with a history of medial tibial stress syndrome exhibit higher plantar-flexion strength in their first MTPJ [11]. Regarding adaptation in sports, this study suggests that it is important to consider the force generation on the hallux and lesser toes separately. Moreover, the composition ratio of the intrinsic and extrinsic muscles is different between the plantar-flexion muscle of the first and second–fifth MTPJs [12]. Consequently, there is a possibility that the sensitivity of the plantar-flexion muscle to the joint angle is different for the first and second fifth MTPJs.

Recently, we developed a device that can measure the plantar-flexion torque of the first MTPJ and a unit of the second–fifth MTPJs at various MTPJ and ankle positions [13]. The purpose of this study was to investigate the difference in the force generation characteristics between the plantar-flexion of the first and second–fifth MTPJs by comparing the MVC torque at different MTPJs and ankle positions using a custom-made device. We hypothesized that the MVC torque of the first MTPJ is influenced by the plantar-flexion/dorsiflexion of the ankle due to the large composition ratio of the extrinsic muscles, whereas the MVC torque of the second–fifth MTPJs is less affected by the ankle position, because of the small composition ratio of the extrinsic muscles.

## 2. Materials and Methods

### 2.1. Subjects

Ten healthy young men voluntarily participated in this study. The mean ± standard deviations of their age, height, and mass were 23.2 ± 1.9 years, 169.2 ± 3.7 cm, and 58.4 ± 8.1 kg, respectively. The required sample size for a repeated analysis of variance (ANOVA) [effect size = 0.25, α error = 0.05, power = 0.80, correlation among repeated measures = 0.6] was calculated using a statistical power analysis software (G*power 3.1, Heinrich Hein University, Düsseldorf, Germany), and the value obtained was 10. Subjects were given precise information about the content of the study, and informed consent was obtained from all subjects. This study was approved by institutional human research ethics committee (approval number: 2013-195) and was carried out in accordance with the declaration of Helsinki.

### 2.2. Measurement of MVC Torque

The MVC torque was measured using a custom-made torque-measuring device. This device recorded the tensile force data from the strain gauge (TU-BR, TEAC, Tokyo, Japan), and converted it from analog to digital using an A/D converter (Power Lab, AD instruments, Bella Vista, NSW, Australia) via an amplifier (DPM-711B, Kyowa Electronics, Tokyo, Japan). The torque values were calculated as the corresponding tensile forces multiplied by the 0.10 m lever arm of the force plate.

Each subject was seated in a dedicated chair and their trunk was secured to the chair using non-elastic straps. Their right foot (dominant side) was secured to the torque-measuring device (Figure 1). To prevent vertical movement during MVC, the lower leg was secured with non-elastic straps. When measuring the MVC torque of the first MTPJ and second–fifth MTPJs, the first MTPJ and second MTPJ were placed on the rotation axis of the force plate. To align the direction of rotation of the MTPJ and force plate, the toes were placed perpendicular to the axis of rotation of the force plate. The position of the foot differed depending on the toes being tested. The toes that were not involved in the measurement of the torque were outside the force plate. Additionally, it was confirmed that the bottom of the toes and foot of the subject did not float from the device in all measurements. After a warming-up session with two subjective 60% contractions without a rest period, each subject performed the MVC torques of the first and the second–fifth MTPJs at each position for approximately 3 s. The results obtained were used to determine the MVC torque. The subjects were verbally instructed to avoid countermovement. The plantar-flexion torque was calculated from the tensile force and lever arm of the foot plate (0.10 m). The MVC torque was calculated as the highest torque minus the lowest torque obtained during contraction (Figure 2). To avoid the effects of muscle fatigue, the resting period was set to at least 2 min. For the analysis, the higher torque between the two measurements conducted at each position was chosen. The reliability of the measurements was previously reported [13].

The MVC torques of the first and second–fifth MTPJs were measured at four MTPJ positions and three ankle positions for a total of 12 positions. The MTPJs were measured at 0°, 15°, 30°, and 45° dorsiflexed positions (0°, DF15°, DF30°, and DF45°, respectively), and the ankle was measured at 20° plantar-flexed position (PF20°), neutral position (0°), and 20° dorsiflexed position (DF20°). These positions were determined to cover the typical joint angle at push-off and the range of motion during running and sprinting [14,15]. The neutral position of the MTPJ was defined as the position parallel to the sole of the foot and toes. The neutral position of the ankle was defined as the perpendicular position between the sole of the foot and the longitudinal axis of the fibula. The ankle position was measured using a goniometer prior to each measurement. The MVC torque was measured twice in each position. For both types of MTPJs, the MVC torque values were measured at four MTPJ positions and at a single ankle position per day. This series of measurements was spread over three days for three different ankle positions. The order of the measured MTPJ and ankle was randomly chosen. The values of the intra-subject coefficient of variation and the standard error of the measurement were presented in Appendix A.

### 2.3. Statistical Analysis

Descriptive data are presented as mean ± SD. Shapiro–Wilk test results were used to verify that all components followed a normal distribution. Therefore, for each MTPJ, two-way repeated measures ANOVA (MTPJ position × ankle position) were used to investigate the interaction or main effect of the MTPJ position and ankle position on the MVC torque. When a significant interaction or main effect was found in the ANOVA, Holm’s multiple comparison test was performed. For all the MTPJs and ankle positions, the MVC torque values between the first and second–fifth MTPJs were compared using the independent-samples t-test. Statistical analyses were performed using a statistical software (SPSS Statistics 26, IBM, Chicago, IL, USA). For all tests, the statistical significance was set as *p* < 0.05.

## 3. Results

For the first MTPJ, there was a significant interaction effect between the MTPJ and ankle positions on the MVC torques in the ANOVA (*p* < 0.01, *F* = 5.05). In multiple comparisons of the effect of the MTPJ positions, when the ankle was at PF20°, the MVC torques of the first MTPJ were significantly different from all other torques and increased as the MTPJ dorsiflexed (Figure 3A). When the ankle was at 0°, the MVC torques increased significantly as the MTPJ dorsiflexed in four comparisons (0° and DF30°, 0° and DF45°, DF15° and DF45° as well as DF30° and DF45°). When the ankle was at DF20°, the MVC torque at 0° of the MTPJ was significantly lower than that at DF15° and DF45°. In contrast, there were no significant differences in the MVC torques at DF15°, DF30°, and DF45°. In multiple comparisons of the effect of the ankle positions, when the first MTPJ was at 0°, DF15°, and DF30°, the MVC torques of the first MTPJ at PF20° of the ankle were smaller than those at DF20° (Figure 3B). In contrast, there was no significant difference between the MVC torques of the first MTPJ at different ankle positions.

For the second–fifth MTPJ, there was no significant interaction effect between the MTPJs and ankle positions on the MVC torques in the ANOVA (*p* = 0.28, *F* = 1.28). However, there was a significant main effect of the MTPJ angles (*p* < 0.01, *F* = 20.43). There was no significant main effect of the ankle angles (*p* = 0.09, *F* = 3.43). In multiple comparisons of the effect of the MTPJ positions, the MVC torques of the second–fifth MTPJs increased significantly as the MTPJs dorsiflexed in five comparisons (0° and DF15°, 0° and DF30°, 0° and DF45°, DF15° and DF45°, as well as DF30° and DF45°) (Figure 4).

The highest torque values obtained for the first and second–fifth MTPJs, were 11.6 ± 2.5 and 8.0 ± 1.8 Nm, respectively. The position at which the highest values of torque were obtained was DF45° of the MTPJ at 0° of the ankle. According to the independent-samples t-test results, the MVC torque of the first MTPJ was found to be larger than that of the second–fifth MTPJs at all positions (*p* < 0.01 for all positions).

## 4. Discussion

This study demonstrated the torque-angle relationships for the first and second–fifth MTPJs. The main finding of the present study was that the force generated on the first MTPJ was affected by the ankle position. To the best of our knowledge, this is the first study to show each force generation characteristic of the first MTPJ and second–fifth MTPJs.

The MVC torques of the first MTPJ increased as the MTPJ was dorsiflexed when the ankle was at PF20° and 0°. However, we observed no significant difference between the MVC torques measured at DF15°, DF30°, and DF45° of the first MTPJ when the ankle was at DF20°. The MVC torques of the second–fifth MTPJs increased as the MTPJ dorsiflexed. The torque-angle relationship is classified into ascending limbs, plateau region (i.e., optimum angle zone), and descending limb [16]. The results of the present study suggest that the torque-angle relationship corresponds to the ascending limb at 0°–DF45° of the first MTPJ, when the ankle was at PF20° and 0° and the plateau region at DF15°–DF45° of the first MTPJ when the ankle was at DF20°. In addition, the torque-angle relationship corresponds to the ascending limb at 0°–45° dorsiflexion of the second–fifth MTPJs when the ankle was at PF20°, 0°, and DF20°. The maximal torque is generated at DF20° of the first MTPJ during sprinting [15]. In addition, the ankle lies in the neutral to plantar-flexed position when maximal torque is generated at the first MTPJ during sprinting [14,17]. The obtained results indicate that plantar-flexion torque of the MTPJs was generated in the ascending limb of the torque-angle relationship during sprinting. In a previous study, the high location of the stiff plate in the shoe induced more MTPJ dorsiflexion than the low location during running; it decreased the torques of the MTPJ and other joints of the leg during running, which could contribute to running performance [18]. Therefore, it may be advantageous not to limit the dorsiflexion of the MTPJ during running and sprinting.

The MVC torque of the first MTPJ at DF20° of the ankle was higher than that at PF20°of the ankle when the first MTPJ was 0°, DF15°, and DF30°. However, there was no significant variation in the MVC torque of the second–fifth MTPJs between the different ankle positions. These results supported our hypothesis. The plantar-flexion moment arm of the ankle is larger in the flexor hallucis longus muscle compared to the flexor digitorum longus muscle [19]. As a result, the flexor hallucis longus muscle varies more in length compared to the flexor digitorum longus muscle during the plantar-flexion/dorsiflexion of the ankle. In addition, the composition ratio of the extrinsic plantar-flexion muscles of the first MTPJ is larger than those of the second–fifth MTPJ [12]. Therefore, we considered that the extrinsic muscles make a large contribution to the plantar-flexion torque of the MTPJ in the dorsiflexion position of the ankle, and the muscle activity of the first MTPJ becomes relatively large at the dorsiflexed position of the ankle. These findings suggest that plantar-flexion of the ankle is needed for the selective exercise of the intrinsic muscle of the hallux; however, exercising the intrinsic and extrinsic muscles of the lesser toes by changing the plantar-flexion/dorsiflexion position of the ankle is difficult. In particular, intrinsic muscles responsible for the plantar-flexion of the hallux (e.g., abductor hallucis and flexor hallucis brevis) are selectively atrophied in patients with pes planus [5]; therefore, plantar-flexion of the hallux at the plantar-flexed position of the ankle could be useful as a strength exercise for such patients.

The highest torques were 11.6 ± 2.5 and 8.0 ± 1.8 Nm on the first and second–fifth MTPJs, respectively. The torques measured in the present study were higher than those estimated in a previous study [12], wherein the productivity of the torque was calculated from the anatomical cross-sectional area and estimated muscle tensions reported in a study of cadavers. However, physiological cross-sectional area has been reported to be more suitable for predicting functional properties than anatomical cross-sectional area [20]. In addition, the force that a muscle can generate per unit area is altered by the number and firing rate of a motor unit [21], and varies from muscle to muscle [22]. Thus, an estimated value may be different from the measured value. Consequently, the in vivo measured torques of this study were higher than the estimated values reported in the previous study. Furthermore, the MVC torque was particularly greater in the first MTPJ in the present study. During walking, humans push off from an axis between the first and second MTPJ [23]. In such cases, the first MTPJ is greatly dorsiflexed [24]. These walking characteristics possibly contribute to the development of the motor unit of the hallux. On the other hand, the MVC torque in the present study was lower than the plantar-flexion torque of the MTPJ during running [25]. A previous study reported that the intrinsic foot muscle lengthens and recoils rapidly during the later stance in accordance with the recoil of the foot arch during running [26]. It is considered that this recoil action causes the stretch shortening cycle [27], which results in an increased torque production.

There are some limitations to this study. First, when generating the isometric force with a finger, the other fingers of the hand also generating a certain force [28], thus, this phenomenon may occur in the toe muscles. To minimize this effects, the unmeasured toes were not placed on the force plate. Second, the second–fifth MTPJs were not aligned perpendicular to the longitudinal axis of the foot. On the other hand, to align the direction of rotation of the MTPJ and force plate, the toes were placed perpendicular to the axis of rotation of the force plate in this study. However, since this study mainly investigated intra-subject factors, these were not considered to affect the results. Third, since the subjects were healthy young men, the applicable range of the results may be limited. Previous studies have shown that the optimal angle for force production is independent of age or gender [29]. Additionally, arch height is not correlated with toe grip strength [30]. However, there is room to investigate the behavior of the torque-angle relationship among the wide population to understand the toe function.

## 5. Conclusions

This study demonstrated the difference in the force generation characteristics of the plantar-flexion between the first and second–fifth MTPJs by comparing the MVC torque of different MTPJ and ankle positions. For the first MTPJ, when the MTPJ was dorsiflexed at 0°–30°, the MVC torque at DF20° of the ankle was higher than that at PF20° of the ankle. However, for the second–fifth MTPJs, there was no significant main effect in the ankle positions. Thus, the present study suggests that force generation on the first MTPJ is more affected by the position of the ankle than that of the second–fifth MTPJs.

## Figures and Tables

**Figure 1 biology-10-00048-f001:**
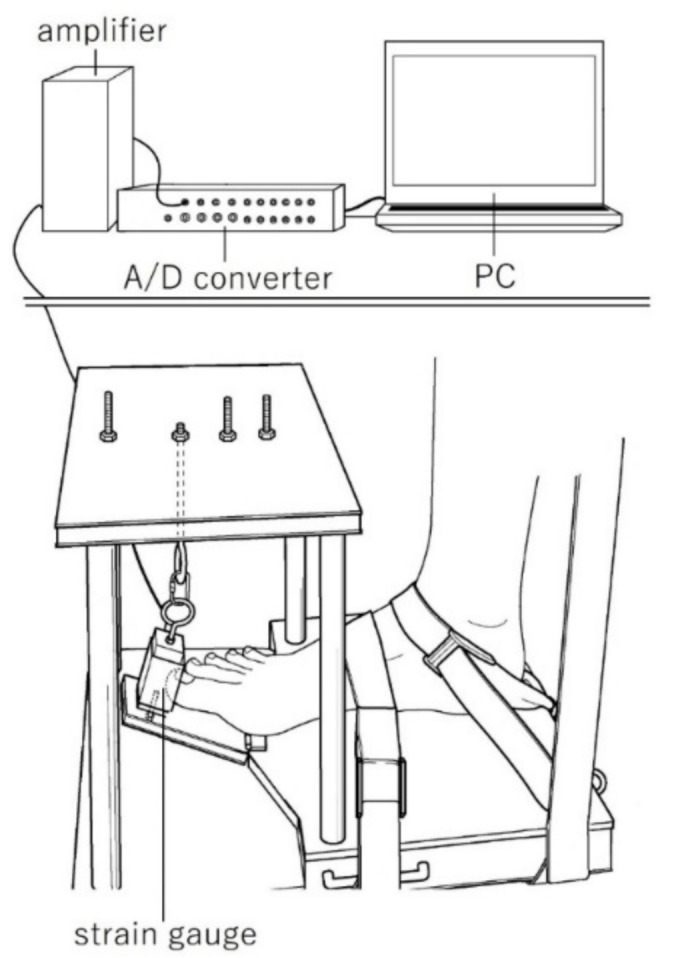
Structure of the metatarsophalangeal joint (MTPJ) plantar-flexion torque-mater for measuring the isometric torque.

**Figure 2 biology-10-00048-f002:**
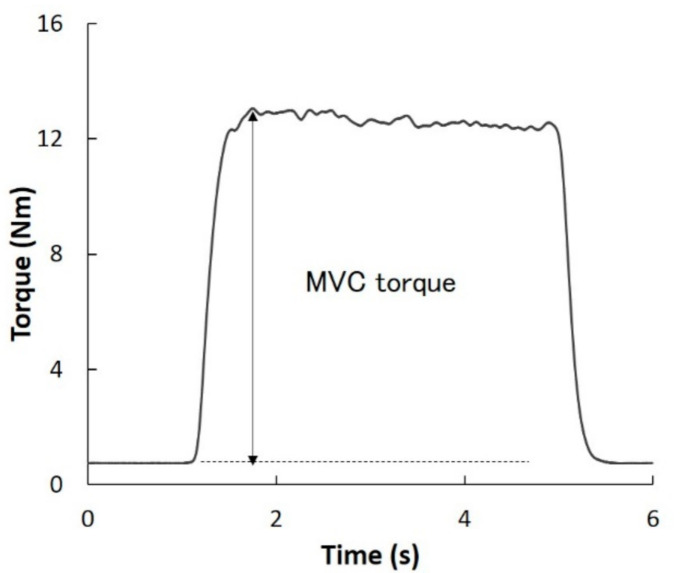
Typical data of the measured maximal voluntary isometric plantar-flexion torque of each metatarsophalangeal joint. Maximal voluntary isometric plantar-flexion torque was defined as the difference between the maximal torque during maximal voluntary isometric contraction (MVC) and passive torque at rest.

**Figure 3 biology-10-00048-f003:**
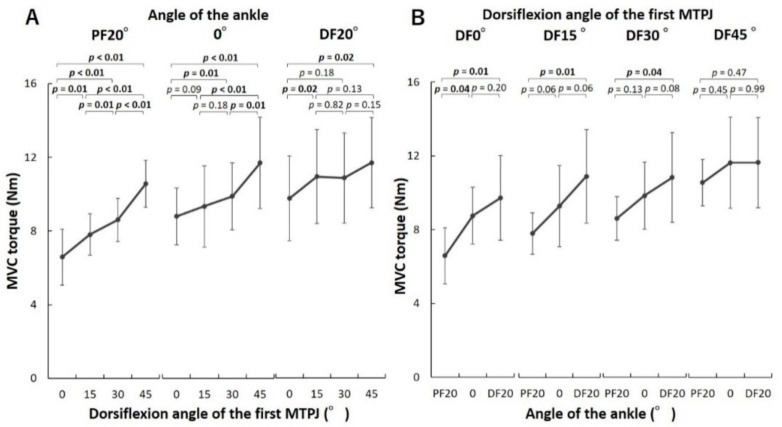
Comparison of the maximal voluntary contraction (MVC) torque of the first metatarsophalangeal joint (MTPJ) at different MTPJ (**A**) and ankle positions (**B**). The torque-angle relationship corresponds to the ascending limb at dorsiflexed (DF) 0°–DF45° of the first MTPJ when the ankle was at plantar-flexed (PF) 20° and neutral (0°), and the plateau region at DF15°–45° of the first MTPJ, when the ankle was at DF20°.

**Figure 4 biology-10-00048-f004:**
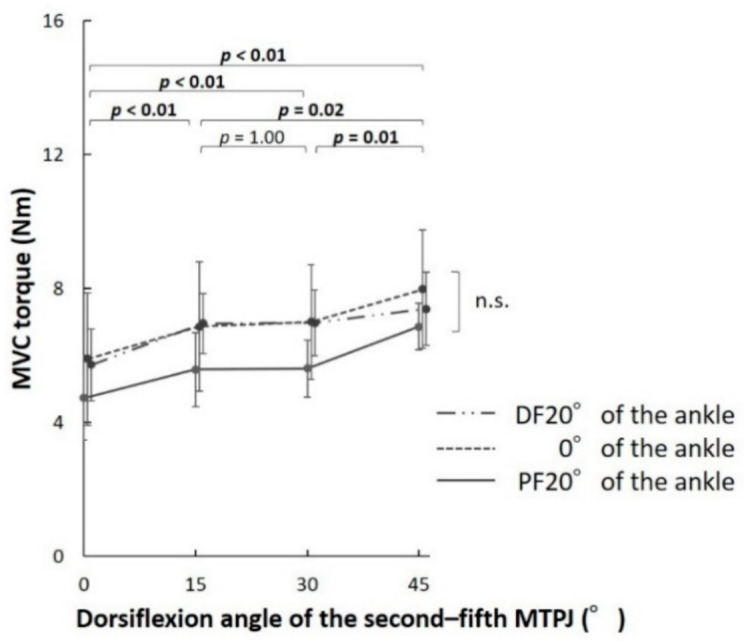
Comparison of the maximal voluntary contraction (MVC) torque of the second–fifth metatarsophalangeal joint (MTPJ) at different MTPJ positions. The torque-angle relationship corresponds to the ascending limb at dorsiflexed (DF) 0°–45°of the second–fifth MTPJ when the ankle was at plantar-flexed (PF) 20°, neutral (0°), and DF20°.

## Data Availability

The data presented in this study are available on request from the corresponding author.

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
