# Peer review of "Force Generation on the Hallux Is More Affected by the Ankle Joint Angle than the Lesser Toes: An In Vivo Human Study"

_biology, 2021, doi:10.3390/biology10010048_

Round 1

Reviewer 1 Report

Updated comments (after reading the completed manuscript)

I find Ok the paper and study. So, I recommend to accept the paper

----------------------------------------

I find the study very interesting, but unfortunately I'm not able to assess completely as tables and figures are not present in the current file. So, the complete data about the sample, or the device used, and also the results are not complete without the tables and figures. 

Author Response

I am very pleased that you are interested in our manuscript. I am very grateful to you for checking the complete data after this comment.

Reviewer 2 Report

I was pleased to review the manuscript entitled: Force generation of the hallux is more sensitive to the ankle and metatarsophalangeal joint angle than the lesser toes.

The aim of the study was to test the hypothesis that force generation characteristics of the hallux and lesser toes are different. Authors tested this in different joint configurations and concluded that the force developed by the hallux was more sensitive to the MTPJ and ankle positions than the second–fifth toes.

The manuscript is built on the premise that hallux and lesser toes are independent entities. However, the authors should acknowledge that some level of enslaving is present among toes so that when asked to maximally contract the plantarflexor muscles of the hallux, the lesser toes will also plantarflex.

The rationale for the study is not clear to me. That the hallux is controlled by bigger muscles and has a more relevant role for stability of the midfoot arch, balance, and propulsion is a well-known fact (Hicks, 1954; Hutton & Dhanendran, 1979, 1981).

I found the discussion to be confusing. The force-length relationship is used to contextualize the results of the study but this has not been measured directly, only inferred. Then the torque-angle relationship is used but not fully explained. For instance, the plateau region (optimal angle zone), the descending and ascending limbs need more explanation. Perhaps in a figure?

I do not see a clear statement of the significance of the study. What are the implications? How your results could be useful? This should be more clearly addressed in the discussion.

Methodology wise, it is not clear where the lesser toes are sitting on the device? Is there an independent plate for the lesser toes or is the position of the foot different depending on the toes being tested?

How the angle has been hold in position during contraction? What is preventing the subject from moving the ankle during contraction? Is there a strap going over the knee? (this is not clear).

Did you account for the inclination angle of the lesser toes? That is, second–fifth MTPJs are not aligned perpendicular to the longitudinal axis of the foot. Was the foot at an angle when the lesser toes were tested?

You should provide more details on the warming-up session. How many contractions? Resting period between contractions?

Average torque between only two measurements may be misleading unless you report the values for each participant on those two contractions. I suggest add this in a table as supplementary material. The point is that unless the two contractions were very similar (i.e. less than 5% difference), the average measure could be unrepresentative of the real ability of the participant for that given condition. The intra-subject variability should be reported for clarity.

Usually, at least three measurements are taken and the two most similar are used for analysis if the difference is less than 5%. See J.-P. Goldmann, Sanno, Willwacher, Heinrich, and Brüggemann (2013)

The choice of the highest single value rather than the average of all values (or part of the values) during contraction is arguable. See J.-P. Goldmann et al. (2013)

Subjects. Table 1 is redundant with information in text. I would suggest remove one or the other.

Figure 3 and 4 have an A and B section but those are not explained in the caption. Please address.

Intro

Line 49-50: Not clear what you mean with “is activated selectively at the knee flexion position”, I think this concept needs to be explained.

Line 50-52: not clear what the message here?

Line 63-65: consider merging these two sentences.

Line 125: the sentence “the Friedman test was used for the ankle and MTPJ angles” does not make sense to me. Statistical analysis section could be more clearly explained. Please state the test, the dependent variables, and the factors tested. For instance:” A General Linear Model for repeated measures was used to compare the main effects of ankle joint and MPJ position on the external moments around the MPJ”. From J. P. Goldmann and Brüggemann (2012)

Line 168-171: these sentences are repeated!

Line 192-193: this concept should be explained more. How is the dorsiflexion of the MTPJ during running and sprinting limited? And how greater torque would help?

Line 223: first sentence is grammatically wrong.

REFERENCES

Goldmann, J.-P., Sanno, M., Willwacher, S., Heinrich, K., & Brüggemann, G.-P. (2013). The potential of toe flexor muscles to enhance performance. Journal of Sports Sciences, 31(4), 424-433.

Goldmann, J. P., & Brüggemann, G. P. (2012). The potential of human toe flexor muscles to produce force. Journal of anatomy, 221(2), 187-194.

Hicks, J. (1954). The mechanics of the foot: II. The plantar aponeurosis and the arch. Journal of anatomy, 88(Pt 1), 25.

Hutton, W., & Dhanendran, M. (1979). A study of the distribution of load under the normal foot during walking. International orthopaedics, 3(2), 153-157.

Hutton, W., & Dhanendran, M. (1981). The mechanics of normal and hallux valgus feet—a quantitative study. Clinical Orthopaedics and Related Research®, 157, 7-13.

Author Response

I am grateful for your reviewing. Please see the attached file.

Reviewer 3 Report

This study aimed to "determine the relationship between the MVC of the plantar-flexion torque and joint angle in the first and second–fifth MTPJs using this device."

This is an interesting study that presents useful data that may impact factor research and, hence, clinical practice. The manuscripts do have some strengths, but I have a few concerns and suggestions which I believe would help improving the manuscript. 

  • First, I think the aim should be re-written. The authors set the aim as "determine a relationship". "Relationship" leads the reader to correlation analysis, which is not the case. Also, while describe the hypothesis, the authors use "is sensitive". From a scientific standpoint, I'm not sure what "sensitive" means here. Please use stronger scientific language.
  • The authors missed to present an hypothesis to 2nd-5th MTPJs.
  • Statistical section is slightly confused. I took a while to get the statistical design. Please make it clear. For example, put in parenthesis how many levels from each factor were in the design. Based on your aims/methods, I would expect a MTPJ Angle x Ankle Angle design. Based on your figures, I'm not sure if that is the case. 
  •  Results section is even more confused, maybe because stats are not Chrystal clear to me. Please reorganized considering your aims; e.g. present if factor by factor
  • I would need to see the stats and results improved to better judge the discussion and how the authors interpreted their findings. Regardless, I suggest the authors to incorporate some functional aspects to the discussion: compare-contrast the muscles involved in the 1st MTPJ and 2nd-5th MTPJ...and how it potentially explain (part) of your findings. 

Minor:

L72. ? appears to be text from the template. Please delete.

L98- why the average and not the higher? If you're assessing MVC, the higher should be used.

L119-120. I appreciate the fact that the authors have previously determined the inter-day ICC. However, it's important to have the standard error of measurement (SEM) to better judge the present study results. This is particularly relevant because the measures were taken in 3 different days. 

Table 1 - Height should be reported in m; Weight should be replaced by "Body Mass" 

Author Response

(The authors gave the same response as above.)

Round 2

Reviewer 1 Report

The paper is metodologically sound and very interesting. Well written and easy to understand. 

Author Response

I am pleased to know that you find our manuscript methodologically sound. I am grateful for your review.

Reviewer 3 Report

I appreciate the point-by-point responses from the authors. Some concerns raised were properly clarified/addressed. However I still have one (real) major concern and one minor:

Major: the statistical design is not appropriate. MTPJ position and Ankle position should not be independently tested. 

Minor: I don't understand the argument regarding CV when determining the MVC. MVC stands for "Maximum". If you are averaging the values of two trials, you are underestimating the real Maximum. Maximum...is maximum.
